# Advances in Capacitive Micromachined Ultrasonic Transducers

**DOI:** 10.3390/mi10020152

**Published:** 2019-02-23

**Authors:** Kevin Brenner, Arif Sanli Ergun, Kamyar Firouzi, Morten Fischer Rasmussen, Quintin Stedman, Butrus (Pierre) Khuri–Yakub

**Affiliations:** 1E.L. Ginzton Lab., Stanford University, Stanford, CA 94305, USA; brennerk@stanford.edu (K.B.); sanli.ergun@gmail.com (A.S.E.); kfirouzi@stanford.edu (K.F.); mofi@stanford.edu (M.F.R.); qstedman@stanford.edu (Q.S.); 2Department of Electrical and Electronics Engineering, TOBB University of Economics and Technology, Ankara 06560, Turkey

**Keywords:** capacitive micromachined ultrasonic transducer (CMUT), acoustics, micromachining, capacitive, transducer, modelling, fabrication

## Abstract

Capacitive micromachined ultrasonic transducer (CMUT) technology has enjoyed rapid development in the last decade. Advancements both in fabrication and integration, coupled with improved modelling, has enabled CMUTs to make their way into mainstream ultrasound imaging systems and find commercial success. In this review paper, we touch upon recent advancements in CMUT technology at all levels of abstraction; modeling, fabrication, integration, and applications. Regarding applications, we discuss future trends for CMUTs and their impact within the broad field of biomedical imaging.

## 1. Introduction

The capacitive micromachined ultrasonic transducer (CMUT) started with an idea to make a better airborne ultrasound transducer operating in the MHz frequency range [1]. Later, a simple underwater experiment showed the huge advantage in bandwidth over piezoelectric transducers and motivated the development of a sealed CMUT for immersion applications [2]. A CMUT consists of a flexible top plate suspended over a gap. Transduction is achieved electrostatically, in contrast with piezoelectric transducers. The merit of the CMUT derives from having a very large electric field in the cavity of the capacitor, a field of the order of $10^{8}$ V/m or higher results in an electro-mechanical coupling coefficient that competes with the best piezoelectric materials. The availability of micro-electro-mechanical-systems (MEMS) technologies makes it possible to realize thin vacuum gaps where such high electric fields can be established with relatively low voltages. Thus, viable devices can be realized and even integrated directly on electronic circuits such as complimentary metal-oxide-semiconductor (CMOS). A further and very important development was the discovery of collapse mode operation of the CMUT. In this mode of operation, the CMUT cells are designed so that part of the top plate is in physical contact with the substrate, yet electrically isolated with a dielectric, during normal operation. The transmit and receive sensitivities of the CMUT are further enhanced thus providing a superior solution for ultrasound transducers [3]. In short, the CMUT is a high electric field device, and if one can control the high electric field from issues like charging and breakdown, then one has an ultrasound transducer with superior bandwidth and sensitivity, amenable for integration with electronics, manufactured using traditional integrated circuits fabrication technologies with all its advantages, and can be made flexible for wrapping around a cylinder or even over human tissue.

In this paper, we will review the various aspects of CMUT technology: theory of operation, fabrication with surface and bulk micromachining, electronic integration methods, characterization, and applications. Beyond this overview, further details in the above-named topics will be left to the references.

## 2. Theory and Modeling of Capacitive Ultrasonic Transduction

A CMUT element typically consists of several cells connected in parallel. Each cell is composed of a flexible top plate (also referred to as top electrode) anchored around its edges. A shallow gap is formed between this flexible top plate and a fixed bottom plate. These two plates are made electrically conductive (partially or completely) to form a capacitor with the gap in-between (Figure 1a), making the CMUT cell a variable capacitor. A CMUT presents a challenging modeling problem as multiple physics are involved in its operation. As with any other MEMS device, at the basic level, the mechanics of the plate needs to be modeled along with the electrodynamics. Moreover, a CMUT interacts with an acoustic medium such as air or water to radiate or sense ultrasound; so, the interaction of the acoustic medium with the CMUT plate also needs to be modeled.

Mechanical systems can be converted into electrical circuits by using the analogy between the mechanical and the electrical domains. One way to implement this analogy is to replace the forces in the mechanical domain by voltage sources and velocities by electrical currents. The models of this type are called equivalent circuit models. This methodology serves as a powerful tool for the analysis of electromechanical systems. Equivalent circuit models have been widely used for design and optimization of variety of transducer technologies such as piezoelectric [4] and an CMUTs [5,6].

Advancement in contemporary computing and equivalent circuits has enabled more detailed two-dimensional and three-dimensional finite element (FE) models to calculate the collapse voltage, output pressure, bandwidth, sensitivity, and crosstalk. Finite element models are designed to solve to the exact coupled-field theory of electrostatics, solid mechanics, and acoustics. A variety of finite element tools have been deployed, including but not limited to, ANSYS, COMSOL, COVENTOR, LS-DYNA, PZFlex, as well as custom-made modeling tools [7].

In the following sub-sections, we first explain the basic electromechanics of CMUTs using a simple parallel plate model. Next, we review the basics of equivalent circuit and finite element modeling.

### 2.1. Basic Electromechanics of CMUTs

To the first order, in one dimension, a CMUT can be modeled by a parallel plate capacitor with a moving top electrode as shown in Figure 2. The mechanics of a parallel plate can be approximated by a mass-spring-damper system model, with a spring constant $k_{p}$, mass constant $m_{p}$, and damping constant $r_{p}$. The medium acoustic impedance is simply modeled using a damper, $r_{m}$, and a mass, $m_{m}$. Under direct current (DC) bias, the top electrode is attracted towards the bottom electrode. At equilibrium, the deflection due to the electrostatic force is counter balanced by the mechanical spring force of the membrane. Assuming the top electrode is displaced by *x*, the capacitance of the parallel plate capacitor is given by
(1)$$c\left(x\right)=\frac{A\epsilon_{o}\epsilon_{r}}{g_{eff}-x}.$$

The effective gap height $g_{eff}$ is given as $g_{eff}=\left(t_{i}+t_{m}\right)/\epsilon_{r}+g_{o}$, where *A* is the area of the top electrode, $\epsilon_{o}$ is the permittivity of vacuum, $\epsilon_{r}$ is the relative permittivity of the insulator and the membrane material (assumed here to be the same), $g_{o}$ is the initial gap distance under zero bias voltage, and $t_{i}$ and $t_{m}$ are the insulator and the membrane thickness, respectively. Note that because of the presence of the oxide layer, the effective gap is different than the physical gap, and thus, the relative permittivity is different than that of Vacuum. The dynamics of a CMUT can be studied via the Newton’s second law, namely,
(2)$$m_{p}\frac{d^{2}x}{dt^{2}}+r_{p}\frac{dx}{dt}+k_{p}x=f_{el}+f_{ac}-p_{o},$$
where $f_{el}$ is the force due to electrical loading, $f_{ac}$ is the force due to acoustic loading, and $p_{o}$ is the atmospheric pressure. $r_{p}$ represents the intrinsic viscoelastic damping of the top plate, which is usually negligible, and thus, we henceforth assume $r_{p}=0$. Both forces can be estimated using the principle of minimum potential energy:(3)$$f_{el}=\frac{-\epsilon_{o}\epsilon_{r}AV{\left(t\right)}^{2}}{2{\left(g_{eff}-x\right)}^{2}},\;f_{ac}=-p(t)A,$$
where p(t) is the acoustic pressure. As seen, this equation is nonlinear in the displacement *x*. For most conventional applications, a CMUT, however, is biased by a large DC voltage ($V_{dc}$), and then modulated through a small AC voltage ($V_{ac}$) in the transmit mode or a small acoustic pressure in the receive mode (leading to induction of a small $V_{ac}$). This assumption serves as a basis for linearization of the electrostatic force at $V_{dc}$ and $x_{dc}$. After expanding the total voltage and displacement as $V=V_{dc}+V_{ac}$ and $x=x_{dc}+x_{ac}$, and dropping the second order terms, the linearized equation of motions becomes:(4)$$\left(m_{p}+m_{m}\right)\frac{d^{2}x_{ac}}{dt^{2}}+r_{m}\frac{dx_{ac}}{dt}+\left(k_{p}-k_{s}\right)x_{ac}=\frac{-\epsilon_{o}\epsilon_{r}AV_{dc}V_{ac}\left(t\right)}{2{\left(g_{eff}-x_{dc}\right)}^{2}},$$
with
(5)$$k_{s}=\frac{\epsilon_{o}\epsilon_{r}A{V_{dc}}^{2}}{{\left(g_{eff}-x_{dc}\right)}^{3}},$$
where $k_{s}$ is known as the spring softening effect, which is a function of the DC bias, and implies the resonance frequency of the CMUT shifts as the DC bias increases. As the top electrode moves closer to the bottom electrode due to the applied voltage, the electrical field increases, and the top electrode displaces further, acting as if the spring constant of the top electrode decreases under the influence of the applied voltage. Also, $m_{m}$ and $r_{m}$ are the effect of acoustic loading, which we shall discuss in-depth later in this section.

The DC components can be calculated by solving $k_{p}x_{dc}=f_{el}-p_{o}$. In absence of the atmospheric pressure, this leads to
(6)$$V_{dc}=\sqrt{\frac{2k_{p}x_{dc}}{\epsilon_{o}\epsilon_{r}A}}\left(g_{eff}-x_{dc}\right),$$
which implies an important phenomenon; if the bias voltage is increased beyond a certain value, the top electrode collapses onto the bottom electrode. This means that the displacement of the top electrode can result in an increase in the electric field to the point where the attractive electrostatic force cannot be balanced by the spring force, resulting in the collapse of the top electrode onto the bottom electrode. Mathematically, this occurs when the gradient of the electrostatic force is larger than the gradient of the mechanical force. One can calculate the collapse voltage by equating the gradient of the electrostatic force to zero:(7)$$V_{coll}=\sqrt{8kg_{eff}^{3}/27\epsilon_{o}\epsilon_{r}A},$$
and thus, it can be seen that displacement at $V_{coll}$ is $g_{eff}/3$.

### 2.2. Small-Signal (Linear) Equivalent Circuit Model

Using the linear model presented above, a CMUT can be considered as a two-port network composed of an electrical domain and a mechanical domain (Figure 3). Such an equivalent circuit is a useful tool that captures the small-signal behavior of the CMUT in one dimension. One can perform a variety of simulations, such as calculating the electrical impedance and the small-signal transmit (TX) and receive (RX) sensitivities as a function of frequency.

In the electrical part, $C_{o}$ is the clamped capacitance of the device at the bias voltage and $-C_{o}$ represents the spring softening capacitance. $V_{s}$ and $R_{s}$ represent the input voltage source and its electric resistance. The mechanical membrane and medium acoustic impedances constitute the mechanical part. $F_{s}$ is the force due to an acoustic pressure source, i.e., $F_{s}=pA$. The two parts are coupled together through an electromechanical transformer, picturing a CMUT as a device that transforms electrical energy to mechanical energy and vice versa. For a parallel plate capacitor, the electric field and transformer ratio are given by
(8)$$E_{o}=\frac{V_{dc}}{g_{eff}-x_{dc}},\;C_{o}=\frac{\epsilon_{o}\epsilon_{r}A}{g_{eff}-x_{dc}},\;n=E_{o}C_{o}.$$

Note also that it is easy to verify $n=\sqrt{k_{s}C_{o}}$. When the transducer is operated in vacuum, the mechanical port of the circuit is short-circuited. For immersion devices, the mechanical port is simply terminated by the radiation impedance. In transmit mode, $F_{s}=0$, and in receive mode, $V_{s}=0$.

The maximum small signal output pressure of the transducer can be easily calculated using the equivalent circuit method. The maximum output pressure is obtained at the resonant frequency of the membrane, where all the reactive elements cancel each other out in the mechanical part of the circuit. At this frequency, the output pressure per volt is simply
(9)$$p_{max}=\frac{n}{A}=\frac{\epsilon_{o}V_{dc}}{{\left(g_{eff}-x_{dc}\right)}^{2}},$$
and for a bias voltage close to the collapse voltage, is $p_{max}=\sqrt{3k\epsilon_{o}/2Ag_{eff}}$. In equivalent circuit modeling, the mechanical impedance of the plate is determined by subjecting the plate to a harmonic loading, semi-analytically using either the classical plate theory [5] or finite element method [8,9], in cases where analytical solutions are either tedious or do not exist. This can be used to derive the impedance of plates of various shapes such as circular, rectangular, or hexagonal geometries, or under different boundary (clamping) conditions.

The medium in which the transducer is operating presents an impedance ($Z_{medium}$) to the transducer that must be included in the small-signal equivalent circuit model (see Figure 3b) and can be considered as $Z_{medium}=Z_{a}A$, where $Z_{a}$ is the characteristic acoustic impedance of the plate as a radiator of sound (not to be confused with mechanical impedance). However, since the transducer is in general a small resonator (compared to the wavelength) $Z_{a}$ can be quite different than that of a plane wave. For a circular piston transducer of radius *a*, within an infinite rigid baffle, this acoustic impedance is given by [10]
(10)$$Z_{a}=Z_{o}\left(1-\frac{J_{1}\left(2ka\right)}{ka}+j\frac{H_{1}\left(2ka\right)}{ka}\right),$$
where $J_{1}$ is the Bessel function of the first kind and the first order and $H_{1}$ is the Struve function of the first order. $Z_{o}$ is the plane wave impedance ($Z_{o}=\rho_{o}c_{o}$), where $\rho_{o}$ and $c_{o}$ are the density and speed of sound in the loading fluid. $k=\omega /c_{o}$ is the wave number. Note that $Z_{a}$ has both real and imaginary parts. The real part contributes to the acoustic damping through $r_{m}$ and is known as the radiation impedance. $r_{m}$ is the mechanism by which a CMUT radiates sound. The imaginary part contributes to the mass term through $m_{m}$, and thereby results in shift in the resonance frequency. In summary, rm=jωARe{Za} and mm=jωAIm{Za}.

Having a complete model, one can now determine the resonance frequency as well as transmit and receive sensitivities. The resonance frequency and fractional bandwidth (FBW) are
(11)$$\omega_{o}=\sqrt{\frac{k_{p}-k_{s}}{m_{p}+m_{m}}},\;FBW=\frac{r_{m}}{\sqrt{\left(m_{p}+m_{m}\right)\left(k_{p}-k_{s}\right)}}.$$

For transmit and receive sensitivities, $S_{TX}$ and $S_{RX}$, it can be shown that
(12)$$S_{TX}=|\frac{P_{out}}{V_{in}}|=S_{TX,max}\Xi \left(\omega \right),\;S_{RX}=|\frac{I_{out}}{P_{in}}|=S_{RX,max}\Xi \left(\omega \right),$$
where $S_{TX,max}=n/A,\;S_{RX,max}=nA/r_{m}$ and
(13)$$\Xi \left(\omega \right)={\left[1+{\left(\frac{\omega }{r_{m}}\left(m_{p}+m_{m}\right)\right)}^{2}{\left(1-\frac{\omega_{o}}{\omega }\right)}^{2}\right]}^{-\frac{1}{2}}.$$

### 2.3. Finite Element Modeling

An equivalent circuit model using lumped parameters serves to provide a good approximation but such an approach has many limitations and may not represent all the underlying physics. Finite element analysis is ideal for analyzing such multi-physics systems. The underlying physics for the CMUT operation can be described using partial differential equations with some boundary conditions. Depending on the shape, geometry, and mode of operation of CMUTs, a diverse array of finite element procedures have been developed over the years. In the finite element method, the simulation domain is divided into many small elements, called finite elements, over which a much simpler function is used to approximate the true solution. A set of linear equations is formulated based on minimizing the error between this approximate solution and the true solution.

Conventional CMUTs have been simulated using finite element analysis as described by [11,12]. Depending on the geometry and structure of the CMUT, Finite element models have been implemented in two-dimensions (2D), two-dimensions with axial symmetry (2D-Axisym.), and three-dimensions (3D) models. In the simplest form, the equations of linear elasticity (both static and dynamic) have been used to capture the mechanics of the vibrating plate. Finite element methods have made it possible to include various mechanical nonlinearities with ease. Perhaps the most prevalent one is geometric nonlinearity (also referred to as stress stiffening) in thin CMUT plates (where the thickness to diameter ratio is less than 5%). Stress stiffening is a nonlinear signature, where the vibrating plate stiffens as it is being deformed due to the second order effects of the strain.

The plate is usually anchored at the edges and is immersed in an acoustic medium (i.e., a domain that is governed by the acoustic wave equation). Often, the acoustic domain is terminated at the outer edges (at a certain distance from the plate) via some form of absorbing boundary conditions (ABC) or a perfectly matched layer (PML). This is usually used in the transmit mode and intended to absorb all out-going waves, and thus, model a CMUT cell pulsating in an infinite half-space. To investigate the operation of a single CMUT cell, generally a spherical absorbing boundary is used to emulate the acoustic radiation condition (Figure 4a). For investigating an array of CMUT cells, it is practically not possible to model all the cells, unless the array has a small number of cells or has a specific type of symmetry. The standard procedure to model an array of CMUT cells is known as the wave-guide model, where a single cell CMUT is modeled, however, with periodic boundary conditions (both in 2D and 3D). The wave-guide model is more subtle in axially symmetric geometries, where symmetry boundary conditions (as opposed to periodic conditions) have been widely adopted to mimic the effect the neighboring cells with a good accuracy [12,13,14] (see Figure 4b). The wave-guide models are also used in the receive mode, in the regimes where the incident wavelength is much larger than the dimensions of the CMUT cell.

Care should be taken, however, when using the wave-guide model. The model shown in Figure 4b generates sound waves in the fluid when the membrane is excited. The field quickly converges into plane waves as it propagates away from the membrane and the wave-fronts become parallel to the membrane plane and absorbing boundary. Below a certain cut-off frequency the fluid wave-guide only supports plane-waves, which is ideal for the purpose of modeling the CMUT cell. However if the frequency is high enough, there are also waves propagating at an oblique angle. The cut-off frequency of the wave-guide is given by $f_{c}=1.22v_{L}/2r_{out}$, where $v_{L}$ is the sound velocity in the fluid and $r_{out}$ is the radius of the wave-guide. Above this frequency, due to the oblique incidence on the absorbing boundary, some part of the incidence wave gets reflected at the absorbing boundary. This results in standing waves in the wave-guide.

Perhaps the most challenging FE modeling of CMUTs is the electrostatic force (or variable capacitance), which is inversely proportional to the deformed gap. As such, different Element technologies have been applied, depending on the Software being used. We elaborate on two here: (1) ANSYS: the electrical ports are added to the membrane by segmenting the gap into many parallel plate capacitors as shown in Figure 5. This approach neglects the fringing fields and assumes that the electric field is always perpendicular to the electrodes. (2) COMSOL: the gap is meshed and modeled by solving the actual equation of electrostatics with a deformed mesh. The last part is necessary as the gap changes in response to the pressure or electrical fields. The coupling between the electric field and vibrating plate is through the electromechanical force that is calculated at the top electrode by the Maxwell’s stress tensor [15].

Using the above model one can perform static, harmonic (small signal frequency domain), and transient (large signal time domain) analyses. For small signal analyses, static calculation is first used to pre-stress the plate prior to the harmonic analysis. The static analysis is also needed to calculate the collapse voltage. Next, a harmonic (frequency domain) analysis is performed to determine various CMUT characteristics in the linear regime.

Recent advancements in FE modeling and computer technologies have also enabled exploring a diverse array of sophisticated designs with the ultimate goal of improving performance measures such as sensitivity and bandwidth, These designs include but are not limited to collapsed mode operation and squeeze-film airborne CMUTs. Each of these developments introduces unique challenges in modeling. For example, for collapsed mode operation the mechanical as well as electrical contacts between the substrate and the vibrating plate should be taken into account. Furthermore, the whole electromechanical operation is nonlinear, leading to a need for time-domain analyses [13,14], which are generally computationally costly and less efficient than frequency analyses. Squeeze-film airborne CMUTs are devices with vented cavities and aim at increasing the bandwidth of airborne CMUTs. Modeling CMUTs with vented cavities is particularly challenging as it involves many different physical phenomena. Namely, the air inside the cavity as well as the vent holes will result in additional loading which needs to be modeled as well. The main challenge for simulating CMUTs with vented cavities lies in modeling the squeeze film, the vent channels, and the interaction of the squeeze film with the acoustic medium, for which one should consider modeling the squeeze film losses using both the Reynolds equation as well as the Navier-Stokes equations [16,17,18].

## 3. Fabrication Technologies

### 3.1. Sacrificial Release Process

The first method developed for fabricating CMUTs was the sacrificial release process [1]. This is a surface micromachining processes where the vacuum gap is created by etching a sacrificial layer between the top plate and the substrate. A vacuum-sealing step allows devices suitable for immersion applications [2].

A typical sacrificial release process is shown in Figure 6. First, a silicon nitride insulator layer is deposited on the wafer, followed by a sacrificial polysilicon layer. The sacrificial layer is patterned to define the post area between cells (a). Then, another layer of sacrificial polysilicon is deposited (b). This second sacrificial layer is used to create channels into the cell for the sacrificial polysilicon etch.

Next, silicon nitride is deposited to form the top plate. Holes are made in the top silicon nitride layer to allow access to the sacrificial etch channels (c). A potassium hydroxide (KOH) wet etch is then used to remove the sacrificial polysilicon layer (d). The KOH enters through the openings in the top silicon nitride layer, then proceeds through the etch channels formed by the second sacrificial polysilicon deposition and into the cells. KOH has high selectivity to polysilicon over silicon nitride, so the silicon nitride structure remains undamaged.

The sacrificial release step leaves a gap, which is vented to the atmosphere. The gap is sealed by performing low-pressure chemical vapor deposition (LPCVD) of silicon nitride. This seals off the narrow etch channels (e). The LPCVD is performed at very low pressure, so the gap is effectively vacuum-filled when it is sealed.

Finally, aluminum is deposited to form the top electrode, electrical contacts, and interconnects (f). Sputtering is usually used in order to get conformal coverage. The substrate wafer is highly doped so that it can be used as the back electrode.

In the sacrificial release process, the gap height is set by the combined thicknesses of the two polysilicon depositions (a) and (b), and the top plate thickness is set by a silicon nitride deposition (c). This can make achieving good uniformity challenging compared to a wafer-bonding process. In addition, roughness in the silicon nitride layer can decrease the effective gap height, causing deviations in device performance from the design values [19]. Also, the plate material can have substantial intrinsic stress, which alters the device properties.

On the other hand, sacrificial release processes have several advantages. They are relatively simple and reliable. They avoid the yield challenges of a wafer bonding step. Additionally, it is possible to design sacrificial release processes with a low maximum processing temperature (250 °C) [20], allowing postprocess CMOS integration [21].

#### Vias and 2D Arrays

Additional steps can be added to the sacrificial release process to allow electrical connectivity from the backside using through-wafer via [22,23]. An example of such a device based on the work of Moini et al. [24] is shown in Figure 7a. Holes for the vias are made through the wafer using deep reactive ion etching (DRIE). The holes are lined with conductive silicon and then filled with undoped polysilicon. Conductive polysilicon is also used to make the bottom electrodes for the CMUTs. Processes like this allow the fabrication of 2D arrays with backside electrical contacts, such as the ring array shown in Figure 7b.

With backside electrical contacts, it is feasible to construct 2D arrays with individually-addressable elements. Rectangular 2D arrays have been fabricated for volumetric imaging [23], and ring arrays have been fabricated for forward-looking imaging on the ends of catheters [24].

### 3.2. Wafer Bonding—Basic Process

In this technique, a combination of surface micromachining and silicon on insulator (SOI) technologies are used to fabricate the CMUT [26]. This wafer bonding process greatly simplifies the fabrication and brings new levels of uniformity and control, especially regarding the plate thickness, which is now defined by the device layer of the SOI wafer.

The wafer bonding fabrication process is illustrated below in Figure 8. The starting point of this process is a prime-grade heavily-doped Si wafer. Care should be taken to ensure the carrier density is high enough in both the prime and SOI wafer to limit depletion during operation. The CMUT gap height is then defined by a thermal oxidation of the prime wafer. The gap dimensions and geometry are then defined, lithographically, and transferred into the oxide with a wet or dry etch down to the Si. Next, a second thermal oxidation is used to passivate the exposed Si and prevent shorting between the bottom of the gap and the conducting plate that will ultimately be formed from the SOI wafer. It should be noted that this oxidation should be as high of quality as possible to avoid hysteresis in the device operation from trap charging or ion drift. At this point, a second lithography step can be used to flatten any bumps that may have formed along the upper edge of the cavity due to increase Si flux [27], which can drop the yield of the SOI bonding. Commonly, a ring pattern (around the gap) and etch can simply remove these bumps. At this point, a critical RCA clean is performed followed by immediate direct bonding of the SOI wafer. During bonding, the oxidized Si wafer is brought in contact with the SOI wafer, at which point a weak van der Waals attraction holds the two wafers together, coupled with weak hydrogen bonding. Another thermal oxidation (≈1100 °C) is then used to covalently bond the two wafers. The next steps involve releasing of the handle and buried oxide (BOX) from the SOI wafer to form the CMUT plates. First, the bulk of the SOI handle is removed using mechanical grinding, typically leaving ≈100 μm depending on the grinding uniformity and control. A wet etch in KOH or tetramethylammonium hydroxide (TMAH) is then used to strip the remaining handle, using the BOX as an etch stop. Finally, the BOX layer is removed by another wet etch in hydrofluoric (HF) acid or its derivative, such as buffered oxide etchant (BOE), this time using the Si device layer (i.e., CMUT plate) as the etch stop. At this point, the CMUTs are physically formed and subsequent steps are related to device isolation and contacts. In this basic process, top-side contacts to the bottom electrode are formed by etching via down to the Si wafer. Metal contacts are then deposited by sputtering both within the via and atop the plate. Finally, a device isolation etch is used to separate the metal pads and etch down through the conducting Si plate electrically isolate the device and/or define the array. As a last step, a low-temperature (compatible with the metallization) oxide or nitride deposition can be used to passivate the sidewall of the plate edge and prevent shorting from surface conduction.

Compared to the sacrificial release process, the wafer bonding process offers better control over the plate thickness and gap height, and much less residual stress in the plate. The main drawback of the wafer bonding process is that the wafer-bonding step, itself, is very sensitive to issues such as surface roughness [28] and cleanliness, and as such, can have a low yield. An additional drawback to thw wafer bonding process is the cost and logistical complexity of procuring suitable SOI wafers. To avoid this, silicon nitride can be deposited on a standard wafer and bonded to the substrate wafer form the top plate [29].

#### 3.2.1. Wafer Bonding—LOCOS Process

The LOCOS (local oxidation of silicon) process is a variant of the wafer-bonding process that allows excellent gap height control and reduced parasitic capacitance when compared to the basic wafer bonding process [30]. The process uses local oxidation, a method used to isolate neighboring MOS transistors in which a silicon nitride mask is used to prevent the diffusion of oxidants to the silicon surface in particular locations so that oxide only grows in specific areas.

A typical LOCOS CMUT structure is shown in Figure 9a. The elevated silicon area in the center of the cells is created using a local oxidation step. This step may be masked with silicon oxide [30] or silicon nitride [31]. A second LOCOS step produces the silicon oxide posts which support the top plate. In this way, the gap height can be made small while keeping the post area thick, unlike the simple wafer bonding process where the gap height and post height are coupled. Gap heights as small as 40 nm have been achieved using this process [30]. In the device shown in Figure 9a, there is no metal layer on the top plate. Instead, highly-doped silicon is used to provide conductivity.

#### 3.2.2. Wafer Bonding—Thick BOX, Pre-Charged

A variety of modifications to the basic process flow for wafer-bonded CMUTs exist, each addressing a specific aspect of performance. Charging is one aspect of CMUT performance that has received considerable attention due to its impact on the device reliability. Specifically, charge trapping and ion drift in the insulating layer within the gap can be responsible for hysteresis in the electrostatic deflection of the plate. In response to such charging, the use of an SOI wafer with a thick BOX layer to form the bottom electrodes has been proposed as one solution [32]. The structure is shown in Figure 9b. The purpose of this is to localize the electrical field, which is the underlying cause of charging, to within the gap by patterning vias through the thick BOX layer. In addition, these backside contacts allow a flat and continuous front face on the transducer array, which improves imaging and simplifies packaging. On the flip side of charging, this can be advantageously exploited with pre-charged CMUTs. Here, charging in the gap insulator is used to electrostatically deflect the CMUT plate and reduce or entirely remove the DC pull in voltage [33].

#### 3.2.3. Wafer Bonding—High-K Insulator

Another approach to suppressing charging involves tailoring the dielectric constant of the insulating layer. Like the introduction of high-K gates in the CMOS industry, high-K insulators can play a role in suppressing charging within the CMUT gap. Specifically, high-K insulators have been shown to suppress the field-emission that occurs between the plate and bottom electrode under high-fields [34]. In parallel to modifications to the dielectric constant, the use of improved deposition techniques, such as atomic layer deposition (ALD) can further suppress charging by reducing the trap density in the insulating layer.

#### 3.2.4. Wafer Bonding—Anodic Bonding and Transparent

A valuable departure from the standard (direct) wafer bonding process is the use of anodic bonding, which can enable CMUTs with low thermal budgets and/or transparency. Anodic bonding is typically used to bond glass to a substrate and is facilitated by ion drift within the glass to form an interfacial bond. The structure is shown in Figure 9c. CMUTs with plates formed through such anodic bonding have been demonstrated with simplified processing, greater tolerance to surface roughness, and high reliability [35]. Moreover, anodic bonding has been employed to improve the transparency of CMUTs to enable coupling with optical measurement techniques [36].

#### 3.2.5. Wafer Bonding—Flexible

In general, electronics are beginning to tap into a new frontier of applications enabled by mechanical flexibility. CMUTs also follow this trend and have experienced interesting developments in flexible devices and arrays, enabling a variety of high-impact applications ranging from conformal patches to swallow-able pills. In general, the approach to flexible CMUT arrays can take two paths; creating the array from entirely flexible components or embedding rigid isolated CMUTs within a flexible medium. These two types of flexible CMUTs are illustrated in Figure 10. Regarding entirely flexible CMUTs, the fabrication typically involves forming the device through micromachining of polymer layers. These polymer layers are supported by a rigid substrate and later released through a chemical etch. One recent demonstration used a combination the polymer SU-8 (as a bulk material) and Parylene-C (to seal the cavity) to fabricate CMUTs on a Si wafer [37]. A thin liftoff layer was then used to release the polymer CMUT array from the Si after fabrication. Another recent approach leveraged a rolling lamination process to form an entirely flexible CMUT array [38].

#### 3.2.6. Wafer Bonding—Bendable Arrays

Regarding the second path towards flexible arrays, rigid CMUTs can be embedded within a flexible medium [39], a variety of recent works have shown progress in forming such arrays by back-filling of mechanically-isolated devices with a polymer layer. For example, recent work on a circular pill-shaped imaging platform has demonstrated a process flow whereby a bonded CMUT array is first fashioned on a Si wafer, then isolated with a through-Si etch, followed by trench filling with polydimethylsiloxane (PDMS) [40]. This process flow for fabricating a pill-shaped device is shown below in Figure 11. Similar to the wafer-bonded process flow, the starting point is a CMUT with the SOI BOX layer removed. Next, a top-side metalization is sputtered. The top-side oxide is then etched to begin the device isolation. A second top-side etch into Si is performed, followed by a nitride passivation. Next, under bump metalization (UBM) is applied to the back-side of the wafer. The device isolation is completed with a backside etch and trench filling with a polymer to form the flexible substrate.

### 3.3. Device Structures to Improve Average Displacement

In a conventional CMUT structure, the edges of the cells are clamped and the spring force arises from the bending of a plate of uniform thickness. This results in an average displacement which is much less than the peak displacement (1/3 in the case of circular cells [41]). A number of device structures have been developed to improve the average displacement by creating plates that move more like pistons.

One approach is to make the plate thicker in the center. A thick center area can be fabricated on the bottom of the top plate using a double wafer bonding process, producing the structure in Figure 9d [42]. Here, two SOIs are bonded together and the device layer of one is used to form the piston while the device layer of the other is used to form the plate. More simply, a mass of a material such as gold can be deposited and patterned on the top of the plate [43].

Another structure called a post CMUT uses compliant posts to provide the spring force rather than the flexural movement of the plate [44]. The structure is shown in Figure 9e. The springs are defined using DRIE and the plate is created by wafer bonding [45].

In addition to improving average displacement, the piston and post CMUTs decouple the compliance of the plate from the mass, which allows greater design flexibility. However, they both present fabrication challenges. In the piston structure, any misalignment of the piston mass can change the dynamic behavior of the cell. Fabrication of post CMUTs with high yield has proved complex and challenging [45].

## 4. Integration of Ultrasonic Transducer Arrays with Electronic Circuits

In ultrasound imaging systems, ultrasound probes typically do not contain any active electronics when the element count is low (<256), and all the front-end electronics is in the imaging system. For CMUT arrays this poses a slight problem in the receive chain because of the typically high electrical input impedance of CMUT array elements. High input impedance combined with long cables result in loss of valuable signal-to-noise ratio (SNR) for the CMUTs. This problem is mitigated by including low noise amplifiers and/or buffers inside the probe, which amplify the signal before driving the cables. Even probes with piezoelectric transducers can benefit from using active electronics. When the element count is high such that the number of channels in the imaging system cannot match the number of elements in the array, the ultrasound probe will definitely contain a certain amount of electronics that would take care of some of the front-end processing and reduce the number of transducer channels to match the number of channels in the imaging system. In this respect, CMUT technology has a distinctive advantage over piezoelectric transducer technology because of the variety of electronic integration possibilities it provides. Figure 12 shows three common approaches to electronic integration applied to CMUT arrays.

A unique integration option for CMUTs had been the monolithic integration of the CMUT array with the electronics in which the CMUT array is built on top of the electronic circuitry directly [46,47,48]. Recently, monolithic integration of piezoelectric micromachined ultrasonic transducers (PMUT) with CMOS electronics have also been achieved by virtue of the same advantage, the compatibility of micromachining processes with CMOS processes [49,50]. Monolithic integration has always been considered the gold standard because of the compactness of the result and the elimination of extra integration steps associated with multi-chip approaches. However, this approach requires high volume production to be economically feasible, which is probably the reason why it did not find commercial traction until low cost, portable high volume ultrasound scanners became possible. (e.g., Butterfly Network Inc., Guilford, CT, USA). Whether a one-dimensional array or a two-dimensional array, monolithic integration provides the best interface (in terms of parasitics) between the CMUT array and the electronics. In monolithic integration, first the electronic circuitry (ASIC) is fabricated using a CMOS technology or similar, then the ASIC wafer is planarized using low temperature deposition and chemical-mechanical polishing (CMP) steps, which is followed by the CMUT fabrication using the sacrificial release process. There has been some success reported on using a low-temperature bonding process for CMUT fabrication as well [51].

Multi-chip integration [23,52] has been used to connect 2D CMUT arrays to the electronics for some time now. In this approach the electronics and the CMUT array are built separately and brought together using a variety of integration options: flip-chip bonding with solder reflow, with gold stud bumps and anisotropic conductive films (ACF) or with thermo-compression bonding. These integration methods, though more practical to apply, are not unique to CMUTs and have been applied to piezoelectric matrix arrays with great success as well [53,54]. For full 2D arrays where the number of individual connections is very high, fanning out interconnects to the sides where conventional interconnect schemes (wire bonding) can be used become impractical. In that case, flip-chip bonding the 2D CMUT array onto the electronics becomes a necessity. The amount of electronic integration that can be incorporated into a CMUT probe varies depending on many parameters, ranging from size to power consumption to the type of application. We will now look into some of these varieties and their associated electronics.

### 4.1. Analog Front End Integration for SNR Improvement

CMUTs typically have higher electrical input impedance and suffer from the parasitic capacitance of the interconnection between the transducer array and the system more than piezoelectric transducer arrays do. To alleviate this problem, a simple solution has been proposed and implemented. An array of low noise preamplifiers is integrated into the probe. The preamplifiers are equipped with active or passive switches that isolate them from the transmit circuitry during high voltage transmit events. After the transmit event the switches change position and the received signals go through the low noise preamplifiers. These amplifiers need not have high amplification but should be low noise and be able to drive the low impedance of the cable and the system front end [55]. Such preamplifiers have already been implemented on standard US front-end chip sets [56]. To accommodate the high input impedance of the CMUT elements, high input impedance preamplifiers can be used. When accompanied with buffers high input impedance preamplifiers reduce signal loss due to the loading of the system cables. However, these preamplifiers are still susceptible to parasitic capacitances between the preamplifier and the transducer element. In case of high parasitic capacitance in the CMUT element and between the CMUT element and the preamplifier, transimpedance amplifiers (aka resistive feedback amplifiers) provide an elegant solution by creating a low impedance node at the preamplifier input which essentially shunts the parasitic capacitance. Both these approaches provide good solutions with some gain and wide bandwidth. Another approach is to use capacitive feedback preamplifiers. These amplifiers are similar to transimpedance amplifiers where the feedback resistor is replaced by a feedback capacitor. Because the impedance of the capacitor drops with frequency, these amplifier show low-pass behavior and are somewhat band-limited. On the other hand, because the feedback resistor is eliminated, capacitive feedback amplifiers show excellent noise performance [57].

### 4.2. Row-Column Addressing

Row-column addressed CMUTs have found a renewed interest in recent years. The row-column addressing allows the fabrication of 2D CMUT arrays relatively easily with reduced number of interconnects [46,58,59]. In this type of array, the bottom electrode is divided into lines (as opposed to making the whole silicon substrate a single electrode which is typically the case for 1D arrays). The top electrode on the other hand is also divided into lines that are perpendicular to the bottom electrode lines. The intersection of these perpendicular lines, which we can refer to as azimuth and elevation lines, constitute an element of the 2D array. Each element in the array is accessed by accessing the corresponding row and column electrodes. Hence, only 2N connections are required, rather than $N^{2}$ to access all the elements. The electronics associated with row-column addressed CMUTs can be somewhat more complicated than the front-end electronics for one-dimensional arrays depending on how they are implemented, but they are a lot simpler than that of fully addressable 2D arrays and produce volumetric images. Hence, row-column addressed 2D CMUT arrays present a very attractive alternative to fully addressable 2D arrays. A row-column addressed 2D array can be used as a synthetic 1D array whose elevation aperture is dynamically changed and electronically scanned in the elevation direction. In this scenario the azimuth lines are connected to the system channels directly whereas the elevation lines are connected to the bias voltage(s) through switches. By electronically turning on and off the switches the elevation aperture is adjusted for each firing [60,61,62]. Another approach is to switch the elevation and azimuth apertures between transmit and receive events while applying Fresnel focusing in the elevation direction, which provides isotropic resolution in azimuth and elevation directions [46]. In such systems, a switch matrix is used to selectively connect each azimuth and elevation line to a system channel, to the bias voltage or to ground. Such a switch matrix can be integrated in any one of the ways described above.

### 4.3. Catheter Based Imaging Systems

CMUTs have catalyzed considerable progress in catheter based imaging applications where array size and channel count are severely limited due to physical constraints such as blood vessel diameter. CMUTs offer elegant solutions to the array size problem. The fabrication flexibility allows manufacturing of small, high frequency, linear, ring shaped, cylindrical forward looking and side looking arrays. Small element size and the long cables associated with catheters warrant a higher level of electronic integration at the catheter tip even for piezoelectric transducers, and only more so for CMUTs. The electronic integration options discussed above along with fabrication simplicity makes CMUTs a better option for catheter based imaging systems. To overcome the parasitic capacitance that the cables present CMUTs are integrated with low noise amplifiers and buffers [21,24,63,64]. The size constraint also limits the number of cables that can run through a catheter. Hence, reducing the number of channels becomes a necessity. Analog multiplexing of the transducer channels and applying synthetic aperture imaging can reduce the number of connections to as low as two [65] with some compromise in the imaging quality and frame rate. A more efficient approach is to integrate TX beamformers and analog multiplexers with the CMUT array at the catheter tip to reduce the cable count without compromising the image quality [48,66,67].

### 4.4. Imaging System on a Chip

Developments in the electronics industry have opened up the way to building ultrasound imaging systems on a chip. Combined with post processing methods to fabricate CMUTs, it is now possible to build single chip ultrasound imaging systems. One caveat in this approach is that ultrasound imaging typically involves a very large dynamic range. Transmit voltages and CMUT DC bias voltages are typically high voltages (≈100 V), whereas received signals are very low voltages. Maintaining a high signal-to-noise ratio and frequency bandwidth is probably the most important objective in the receive chain. In addition, there is the digital component of the imaging system. Combining all of these in a single chip provides the best signal integrity but may compromise signal quality to some degree. However, HV-BCD processes seems to overcome such problems [57,67,68]. Building ultrasound imaging systems on a chip or even on multiple chips paves the way to very low cost, ultra portable ultrasound imaging systems. When combined with today’s wireless communication capabilities and battery power technologies, such ultrasound systems turn into flexible, wearable, ingestible standalone ultrasound imaging systems. In our group we have developed and demonstrated the components of such a system which would go into an ingestible capsule form and provide the images of the gastrointestinal tract. This capsule combines an ultrasound imaging system with on-chip transmit and receive beamforming capability, with power management and wireless transfer circuitry [69,70]. The system acquires US images of the intestinal cross-section and wirelessly transmits them to a host computer Figure 13. In addition to the high dynamic range requirements, the form factor of an ingestible US imaging system forces a multichip approach for this integration problem.

### 4.5. Imaging and HIFU System Integration

CMUTs have two very distinctive advantages when it comes to high intensity focused ultrasound (HIFU) applications. One of them is the ability to do imaging and HIFU with the same transducer array given the inherent wide frequency band of the CMUT transducers. The other one is the lack of a self-heating mechanism in CMUTs (whereas for piezoelectric transducers self heating is the main issue). Therefore, CMUT transducer arrays are excellent candidates for dual mode operation. Dual mode operation usually requires specific electronic circuitry that interfaces the HIFU drivers with the transducer arrays. In HIFU operation long bursts of sinusoidal signals are used to excite the elements. The pulsers in the imaging circuitry are usually not efficient enough to generate such kind of long bursts without overheating, and therefore separate HIFU drivers are needed for dual mode operation. The HIFU driver is interfaced to the CMUT array and the imaging system with a switch matrix which switches from imaging to HIFU and back to imaging in a programmable manner to be able to ablate tissue and monitor the results in real time. In our group we have successfully developed and demonstrated such a dual mode imaging/HIFU system for ablation in shallow tissue that uses a 2D CMUT array which is flip-chip bonded to the electronic circuitry [71,72]. The 2D CMUT array consists of dedicated transmit and receive elements. The transmit elements are connected to a TX beamformer whereas the receive elements are connected to low noise receive amplifiers. To reduce power consumption in the ASIC, an off-chip, standalone, eight-channel HIFU driver is used. The ASIC includes high-voltage switches to switch the TX elements between imaging pulsers and HIFU channels Figure 14. With this integrated device, it is possible to do HIFU (up to 16 MPa peak to peak pressure at 1 cm focal depth at 5 MHz) and imaging with the same 2D array. As the level of electronic integration increases thermal management of the transducer array and electronics assembly start to become an important issue, especially in dual mode systems. For example, even though off-chip HIFU drivers are used the amount of power dissipation in the high-voltage switches in the ASIC is enough to increase the probe temperature to unacceptable levels (42 °C US probes). However, the thermal energy levels involved are modest and is easily managed by basic thermal design considerations.

## 5. Applications

As there have already been many applications of CMUTs, we cannot present them all here. Instead, we have selected applications of particular interest from our own work.

### 5.1. Medical Ultrasound Imaging

Medical ultrasound imaging is non-ionizing, safe, and therefore broadly accepted for diagnostic use even for obstetrical applications. Compared to all other imaging modalities, ultrasound imaging is cheap and portable. These features have made ultrasound imaging a widely used imaging modality in medicine. We postulate that CMUTs can replace piezoelectric transducers in any medical ultrasound probe. Butterfly Network has recently released their first probe which is based on a 9000 element 2D CMUT array. The handheld device plugs directly into a smartphone and emulates both a phased array, convex array and linear array. A fetal scan carried out with their probe connected to a smartphone is shown in Figure 15.

### 5.2. Dual-Mode: HIFU and Imaging with the Same Transducer Array

High-intensity focused ultrasound (HIFU) is a minimally invasive, FDA-approved procedure to ablate tissue without damaging the surrounding tissue especially closer to the transducer [74]. Well-known examples of HIFU include targeted ultrasound drug delivery, treatment of uterine fibroids, neurological disorders, prostate cancer, and other types of cancers [75,76]. For a successful HIFU treatment it is required to guide the procedure with an imaging modality like ultrasound imaging or MRI. Normally both imaging modalities are separate from the HIFU system which makes it an engineering task to design a system that guarantees co-registration between the HIFU delivery location with the imaging. Here we present a device where we have combined the imaging and HIFU system into one integrated circuit (IC) that operates on one 32 × 32 element 2D CMUT array and the co-registration is therefore an integral part of the system. The IC can switch between HIFU mode and imaging mode on a millisecond scale giving the illusion to the user that the HIFU ablation and imaging is taking place at the same time. The switching time is limited by the speed of the DC power supplies. The dual-mode device is shown in Figure 16.

### 5.3. Catheter Based Ultrasound Imaging

An alternative to an imaging and HIFU system that operates as one, is to have a system that eases the coupling between the two systems and therefore also the co-registration. We present here a CMUT ring array that has been fully integrated into a catheter assembly with a 12.5 mm OD and 4 mm ID. The circular catheter form-factor makes it easy to couple and co-register the imaging array with a center-piece, see Figure 17a, HIFU system or optical fibers for photoacoustic imaging. The catheter is shown in Figure 17.

## 6. Industrialization

Given the obvious advantage of manufacturing CMUTs using silicon integrated circuits processing technologies, commercialization was considered to be a short distance away from invention. In retrospect, it has been a fairly rapid path to full industrial products and volume production. The following will mention a subset of the industrialization effort with no intention of short-changing efforts that are unknown to the authors.

The first foray into industrialization was Sensant which was founded in 1998 and was eventually acquired by Siemens in 2005. Sensant started trying to commercialize CMUTs for flow gas metering, and eventually got into medical imaging. However, no product ever reached the market out of this effort. In 1998 the ACULAB laboratory at University of Roma Tre started an effort in CMUTs and is providing their technology for commercialization.

Hitachi Medico succeeded in the development and put into practical use a probe named “Mappie” that was attached to a system for detecting breast cancer. The system was introduced in 2008 at the 82nd Annual Scientific meeting of the Japan Society of Ultrasonic in Medicine at Tokyo, Japan. In 2017 Hitachi announced an updated system at the RSNA meeting in Chicago and introduced the product in the US market.

Around the same time frame of 2008, Vermon in Tours, France, introduced CMUT arrays for research purposes using the sacrificial release technology. These arrays were not available in the US till much later in time. In 2005, Kolo Technologies was formed and has continued in developing technology to commercialize CMUTs. Today, Kolo Medical (an outgrowth of Kolo Technologies) specializes in high frequency CMUT arrays (>25 MHz), and has a strong presence in the US and Suzhou, China.

Butterfly Network inc., a seven years old start-up, has made the most impact in successful commercialization of CMUTs. In the Fall of 2018, Butterfly announced the introduction of a handheld probe that plugs to an iPhone to make the full ultrasound imaging system. The system sells for less than $2000 and is FDA approved for 13 different examinations.

At the most recent IEEE Ultrasonics Symposium in Kobe, Japan, Philips announced that it is making available design and fabrication capability of CMUTs (operated in collapse mode) for end users. Philips has been working on CMUTs for a number of years and possesses an excellent technology base.

Similarly, the Fraunhofer Institute for Photonoic Microsystems (IPMS) has been working on CMUTs and is commercializing CMUTs for a number of applications both in the medical space and in airborne ultrasound. IPMS has a previous history in making vibrating ribbons for light modulation (silicon light valve), a device that is very similar to the CMUT in design and construction.

Another player in the space is Canon, which has been working on 3D real time photoacoustic imaging systems for breast screening using 2D arrays of CMUTs. It is expected that they will be on the market with such systems once they receive FDA approval.

Overall, it is estimated that there is a total of about 23 companies offering CMUT products.

## 7. Discussion

Since its inception 25 years ago, CMUT technology have seen significant developments, especially in the last decade. With the advance of the modern modeling tools, both analytical and numerical CMUT models have become accurate and reliable enough to make industrial quality designs possible both for normal mode and collapse mode of operation. Fabrication processes and electronic integration tools have also matured to the point that mass production of CMUT probes along with its accompanying electronics have become economically feasible. As a result, more and more ultrasound imaging companies have started to incorporate CMUT probes into their systems. The ease of miniaturization and variety of electronic integration options enabled low-cost, ultra-portable, wireless or with minimal wiring, wearable or ingestible complete ultrasound imaging systems possible.

CMUTs show distinctive advantages when it comes to high intensity focused ultrasound applications. The lack of self-heating mechanism and inherent wide frequency bandwidth make CMUTs an excellent candidate for HIFU and dual mode imaging/HIFU applications. Although this option hasn’t seen a commercial application yet, primarily because it is an area that is being explored only recently, there has been successful demonstrations of dual mode operation with CMUTs. The future work on CMUTs is likely to include its therapeutic uses as well. Some imaging systems have already started including HIFU capability which should find commercial success in the coming years.

## Figures and Tables

**Figure 1 micromachines-10-00152-f001:**
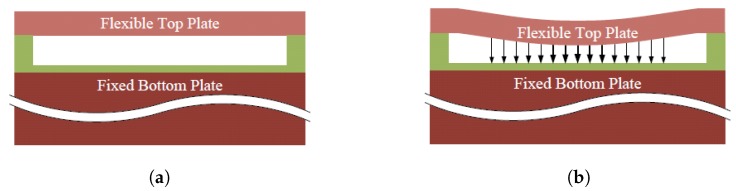
Capacitive micromachined ultrasonic transducer (CMUT) cell illustration. (**a**) A CMUT cell is composed of a flexible top plate and a fixed bottom plate. (**b**) A direct current (DC) bias is applied during the operation that deflects the top plate.

**Figure 2 micromachines-10-00152-f002:**
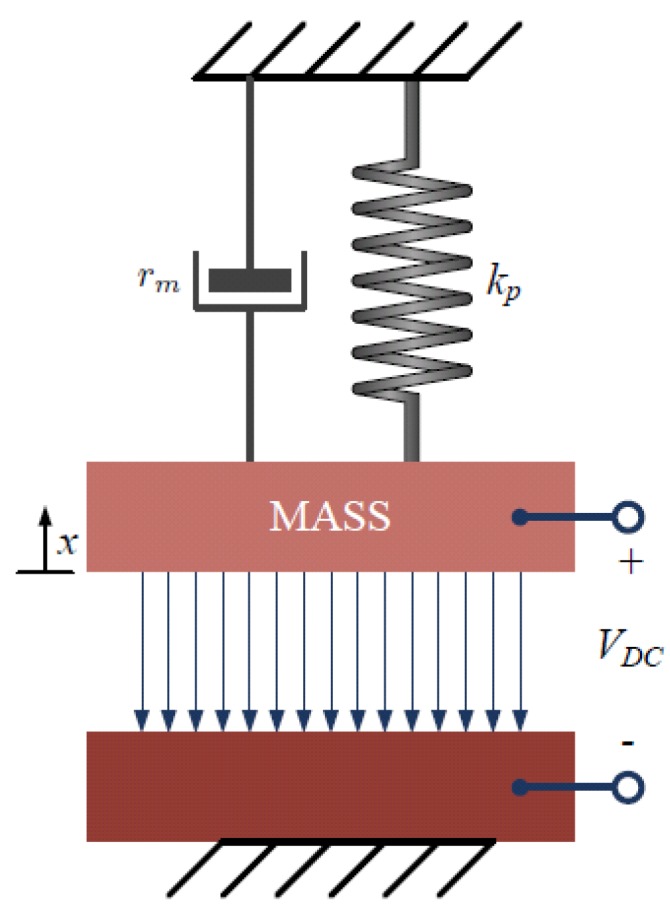
Simplified mass-spring-damper CMUT model.

**Figure 3 micromachines-10-00152-f003:**
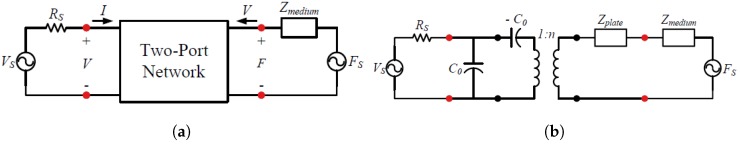
CMUT Network model. (**a**) General two-port network representation that relates voltage and current (*V* and *I*) to force and velocity (*F* and *V*). (**b**) Small-signal equivalent circuit model (in transmit mode, $F_{s}=0$, and in receive mode, $V_{s}=0$). $R_{s}$ represents the electric source resistance.

**Figure 4 micromachines-10-00152-f004:**
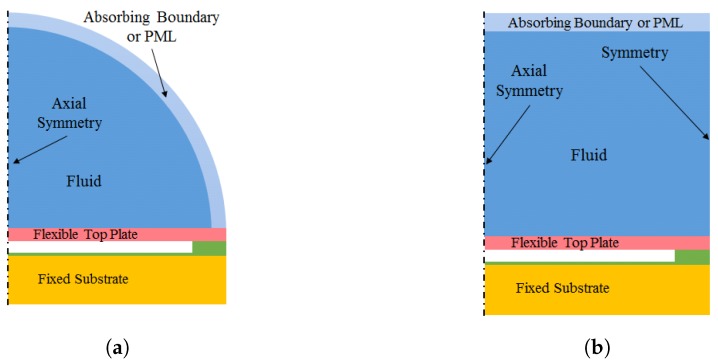
Geometry and boundary conditions of 2D-Axisym. CMUT finite element (FE) models. (**a**) Single cell model. (**b**) Wave-guide model.

**Figure 5 micromachines-10-00152-f005:**
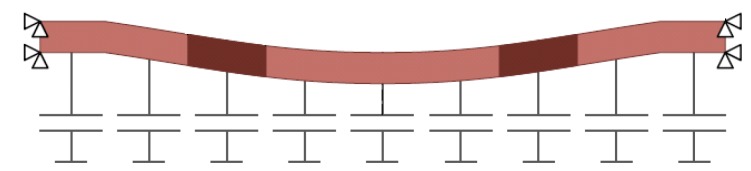
CMUT electrostatic gap segmentation. The electrostatic force is approximated by adding several parallel plate capacitors between the top and bottom electrodes.

**Figure 6 micromachines-10-00152-f006:**
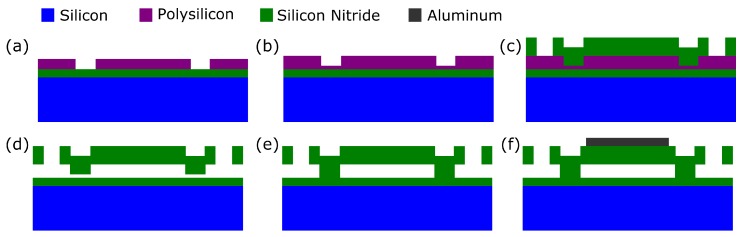
Sacrificial release CMUT fabrication process. (**a**) Deposit silicon nitride. Deposit and pattern the sacrificial polysilicon layer. (**b**) Deposit the polysilicon sacrificial layer for the etch channels into the cells. (**c**) Deposit and pattern the silicon nitride top plate layer. (**d**) Etch the sacrificial polysilicon layer. (**e**) Deposit silicon nitride using low-pressure chemical vapor deposition (LPCVD) to seal the cells. (**f**) Deposit and pattern the aluminum electrodes and interconnects.

**Figure 7 micromachines-10-00152-f007:**
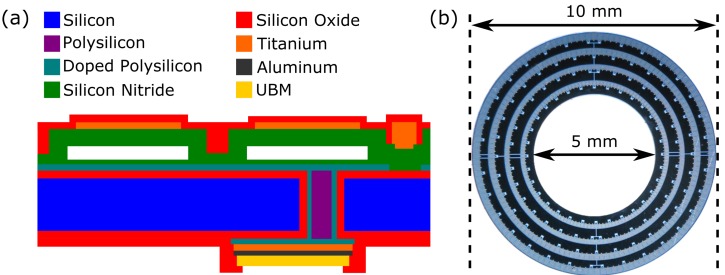
(**a**) Structure of a sacrificial release CMUT with vias for backside electrical contacts. (**b**) A four-ring 2D CMUT array fabricated using a sacrificial release process and through-wafer vias, as decribed in [25]. Reproduced with permission from Moini, A., Capacitive Micromachined Ultrasonic Transducer (CMUT) Arrays for Endoscopic Ultrasound; published by Stanford University, 2016.

**Figure 8 micromachines-10-00152-f008:**
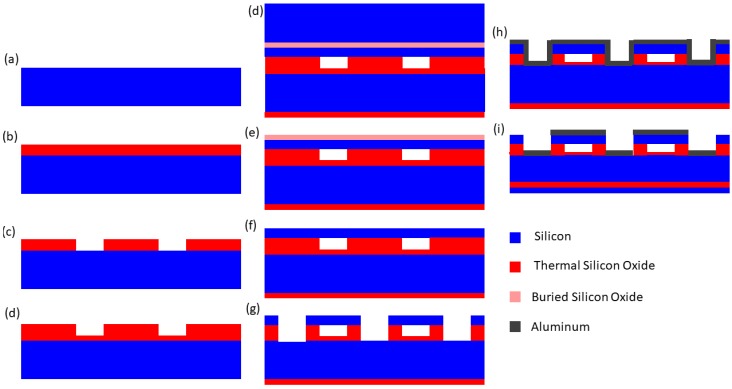
Process flow for a wafer-bonded CMUT. (**a**) Starting prime wafer. (**b**) Thermal oxidation. (**c**) Etch to form cavity. (**d**) Thermal oxidation. (**e**) Silicon on insulator (SOI) wafer bonding. (**f**) SOI handle. (**g**) Removing burried oxide. (**h**) Sputtering metallization. (**i**) Metal pattern and device isolation.

**Figure 9 micromachines-10-00152-f009:**
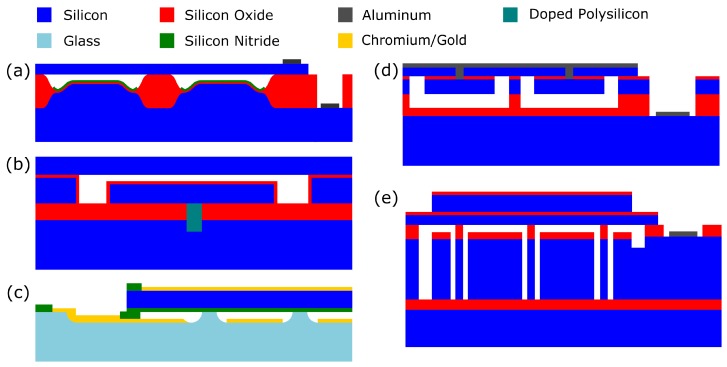
(**a**–**c**) Wafer-bonded CMUT structures. (**a**) Local oxidation of silicon (LOCOS) wafer-bonded CMUT. (**b**) Thick buried oxide (BOX) CMUT. (**c**) Anodic-bonded CMUT. (**d**–**e**) Structures to improve average displacement shown for comparison. (**d**) Piston CMUT fabricated with a double wafer-bonding process. (**e**) Post CMUT.

**Figure 10 micromachines-10-00152-f010:**
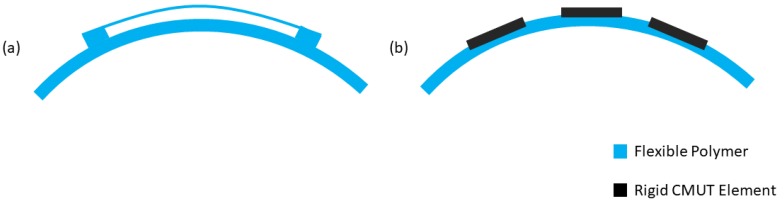
Approaches to fabricating flexible CMUTs. (**a**) CMUT fabricated from entirely flexible polymers. (**b**) Isolated rigid CMUTs embedded within a flexible substrate.

**Figure 11 micromachines-10-00152-f011:**
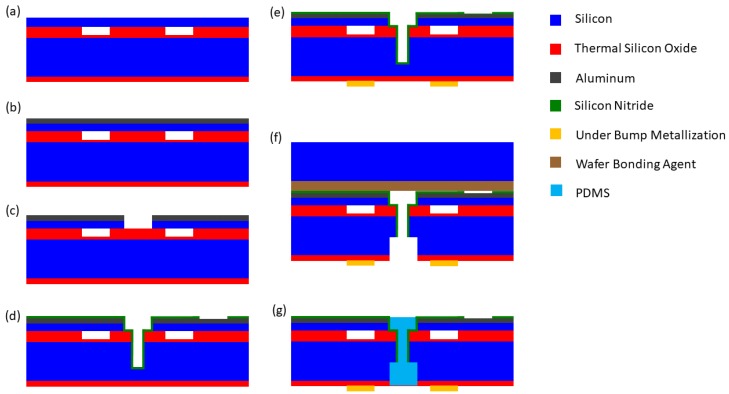
Process flow for fabricating a bendable CMUT array. (**a**) A wafer-bonded CMUT after stripping the SOI BOX. (**b**) Sputtering top-side metallization. (**c**) Top-side isolation etch. (**d**) Second top-side isolation with nitride passivation. (**e**) Deposit of under bump metalization (UBM). (**f**) Secure to supporting wafer. (**g**) Back-side isolation and polydimethylsiloxane (PDMS) filling.

**Figure 12 micromachines-10-00152-f012:**
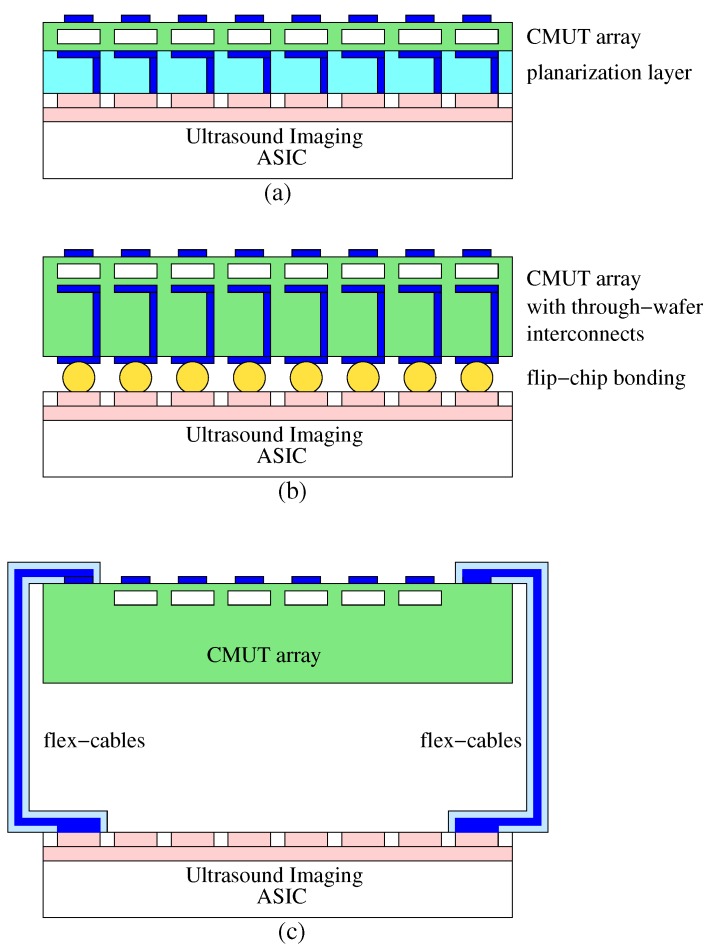
Schemes for electronic integration: (**a**) monolithic integration, (**b**) multi-chip integration, (**c**) hybrid integration.

**Figure 13 micromachines-10-00152-f013:**
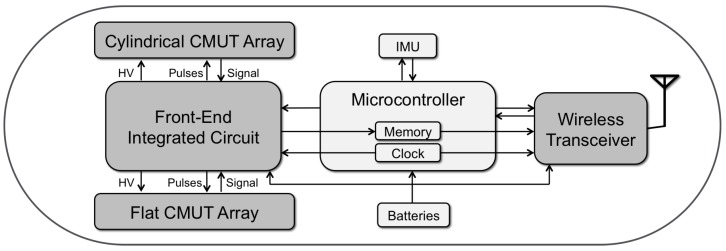
Capsule US imaging system.

**Figure 14 micromachines-10-00152-f014:**
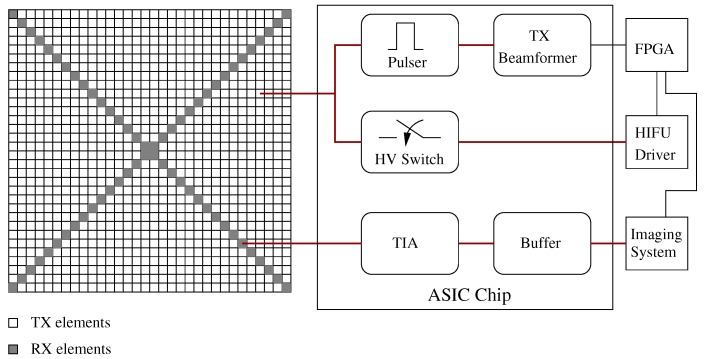
Dual-mode high intensity focused ultrasound (HIFU) and imaging system with 2D CMUT array and electronic circuitry (ASIC).

**Figure 15 micromachines-10-00152-f015:**
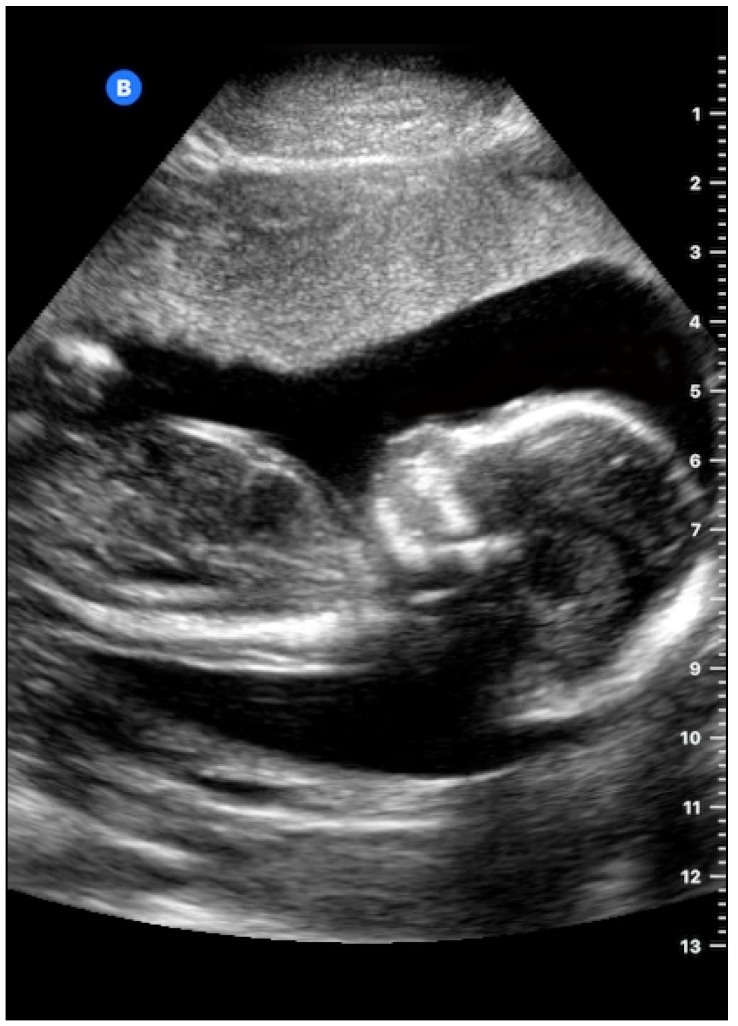
A fetal B-mode scanned with a CMUT transducer reproduced with permission from Butterfly Network [73].

**Figure 16 micromachines-10-00152-f016:**
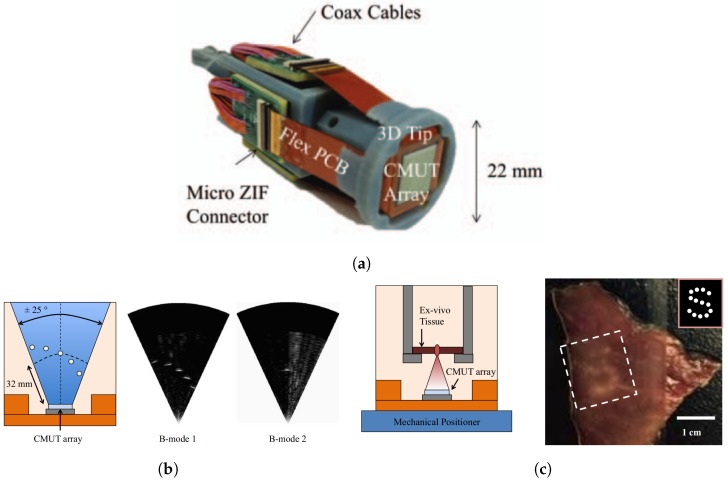
(**a**) The integrated CMUT, IC and flexible printed circuit board (flex PCB) prior to PDMS casting. (**b**) 3D imaging result of a wire phantom. (**c**) HIFU ablation of a piece of ex-vivo tissue. All images from [71]. Reproduced with permission from Jang, J.H. et al., 2015 IEEE International Ultrasonics Symposium (IUS); published by IEEE Xplore, 2015 [72]. Reproduced with permission from Jang, J.H. et al., 2017 IEEE International Ultrasonics Symposium (IUS); published by IEEE Xplore, 2017.

**Figure 17 micromachines-10-00152-f017:**
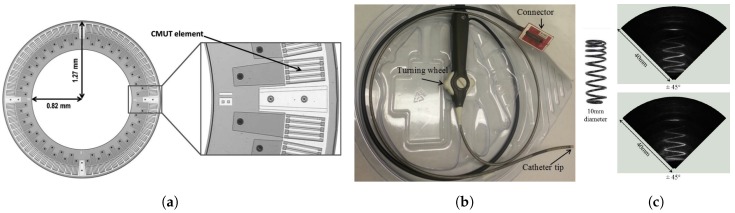
(**a**) CMUT ring array. (**b**) CMUT ring array integrated into a endoscopic assembly. (**c**) 3D imaging results of a spring. Images from [24]. Reproduced with permission from Moini, A. et al., 2016 IEEE International Ultrasonics Symposium (IUS); published by IEEE Xplore, 2016 [66]. Reproduced with permission from Moini, A. et al., ASME 2015 International Technical Conference and Exhibition on Packaging and Integration of Electronic and Photonic Microsystems collocated with the ASME 2015 13th International Conference on Nanochannels, Microchannels, and Minichannels; published by ASME, 2015 [77]. Reproduced with permission from Choe, J.W. et al., IEEE Transactions on Ultrasonics, Ferroelectrics, and Frequency Control; published by IEEE Xplore, 2012.

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
