# Peer review of "Advances in Capacitive Micromachined Ultrasonic Transducers"

_micromachines, 2019, doi:10.3390/mi10020152_

Round 1
Reviewer 1 Report
This is a timely and valuable contribution to the field of CMUT research in providing a useful overview for readers wishing to gain an initial understanding of the many interesting aspects of these devices.
The combination of an overview of modelling, with fabrication, electronics integration, applications and industrial uptake / commercial supply is compelling and well-balanced to provide an accessible source of information for readers who will wish to read more deeply later.
Occasionally the authors let their enthusiasm for the topic override the necessary scientific excellence of their expression of the situation. This should be corrected according to the detailed notes below.
Further, with ultrasound expanding quite rapidly (CAGR > 10% predicted for the next five years) and handheld systems of particular interest (market forecast to expand by a factor x5 in the next five years), the reader will benefit very much from some limited additional text dealing with adjacent technologies such as PMUTs. Again, this is detailed below.
As a final general comment, the paper is generally written in an exemplary manner but there are a few typos and colloquial expressions which are not detailed here.
Detailed comments are as follows:
p1 l4 – Delete “exciting”; this is a word for project proposals not for scientific papers. The reader may decide what is exciting.
p1 l10 – 30 - The introduction reads like a marketing document for CMUTs. It should be rewritten in a more scientifically-appropriate style. Further, given that CMUT technology has been under development for more than 20 years and that the vast majority of ultrasonic transducers remain driven by the piezoelectric phenomenon, it is essential to include balancing text so that the reader may understand properly that CMUTs are an interesting technology for the future, with many advantages, but are still in minority use because of disadvantages.
In this regard, and as a comment that may be included in the text at the authors choice, piezoelectric single crystal technology has gained very widespread adoption in contemporary ultrasonic transducers within an almost identical period of time, whereas CMUT technology is still lagging. It may be helpful to explore parallels.
p1 l22 – It is hard to understand the technical analogy between the collapse mode of a CMUT and turbocharging of an internal combustion engine. The authors should provide an analogy that is easier to understand.
p1 l23 – The authors should briefly explain how shorting of the electric field between the upper and lower electrodes is avoided in the collapse-mode of a CMUT.
p1 l25 – The authors should outline the technical bases in which CMUT technology is “superior” and relate this to a lack of adoption to date.
p1 l29 – 30 – The last sentence in this paragraph should be removed. Please see previous comments here.
p2 l50 – “Piezoelectric” should not have an upper-case “p”
p2 l51 – With the advancement in contemporary computing, equivalent circuit models have less advantage over finite element analysis than was the case historically. Hence the phrase “great success” does not apply contemporaneously; the models remain useful for very rapid calculation but FEA is essential for modeling at the present time and this should be clarified.
As a general point, “finite element” does not have to be capitalized. Please change all text to “finite element”.
p2 l55 – Substitute “meant” with “designed”
p2 l61 – Insert “in one dimension” after “To the first order”. The analysis which follows does not include e.g. the curvature of the upper electrode of a CMUT; this should be made clear.
p2 l63 – 64 – Insert “acoustic” into “medium [acoustic] impedance”. Throughout the paper, please use “acoustic” where acoustic impedance is referred to otherwise readers may be confused, rather than assuming that it will be obvious from context (as it generally is to an established expert – this paper is clearly not written for such experts).
p2 l64 – 67 – At the relatively introductory level of this paper (where its publication will be very valuable) the relationship between the upper “electrode” and the “membrane” (assumed to be of the same permittivity as the insulator on p3) may not be clear to the reader. This should be detailed.
p3 3rd line – It was previously noted that the CMUT uses air (or vacuum) in its cavity. Hence, the introduction of an insulator with a relative permittivity presumably <> 1 will puzzle the non-expert. More detail should be provided.
p3 2nd line after Eqn (2) – r<sub>p is noted as applying to “the plate”. Is this the top or bottom plate, or both? (The top plate is also called the membrane.)
Eqn 3 – In Eqn 3, x is given as plain x, but it is defined as a function of t below. It would be helpful to clarify the symbolism, probably best by dropping the explicit reference to x(t) and simply stating “in the displacement x as a function of time”. (It is always difficult to make an introductory paper completely correct without also making it over-long.)
p4 line above Eqn 7 – “electrostatic force”
p4 l66 – Explain the use of the bar above x (i.e. why is displacement now labelled as x bar?)
p4 l70 – “the small-signal behavior of the CMUT in one dimension”
p4 2nd line above Fig. 3 – define “TX” and “RX”
p4 2nd line below Fig. 3 – Refer to R<sub>s as “resistance” if that is what is meant or substitute with Z<sub>s
p4 2nd line above Eqn 8 – Explain what is meant by an “electromechanical transformer”
Eqn 8 – As stated, this suggests that V<sub>dc = eps<sub>0 eps<sub>R A. However, the units of these quantities do not match. Please clarify.
p5 l73 – It is not obvious how “equivalent circuit modeling” can use results from the “Finite Element method”. Please explain why it is not better just to use a complete finite element model.
p5 3rd line above Eqn 10 – Rephrase “Z<sub>a can be quite different than that of a plane wave” i.e. clarify how Z is related to a plane wave.
p5 l79 – The explanation of radiation impedance is useful but is not in the right place as there is a reference to radiation impedance on p4 l70. Please explain radiation impedance where it is first referenced on p4.
p5 Eqn after Eqn 12 – This eqn is not numbered. Also c<sub>ac is not defined.
p6 l97 – “Finite Element technology” is a strange term. Please use “finite element analysis” or “techniques” or “the finite element method”.
p6 l107 – 108 – The phrase “a spherical boundary is used to emulate the acoustic radiation” is puzzling. (Radiation in this context is a wave phenomenon; a boundary is a physical entity.) Please rephrase more clearly.
p6 l109 – Please explain why it is not possible to model all the cells in an array and note that this is a disadvantage compared with other forms of ultrasonic transducer such as piezoelectric arrays, where the homogenous nature of the ultrasonic source / receiver makes modelling easier.
p6 l132 – Please indicate the degree of approximation implied by neglecting fringing fields. Is it 1%, 5%, 10%?
p7 l145 – “airborne CMUTs” are presumably used in applications such as aircraft, spacecraft and drones. It is not obvious why this changes the modelling that is needed. Is it to do with the speed at which the CMUT is moving? Please also explain why a different design is needed for CMUTs in this particular environment.
p8 Fig 6 – Remove bracket from end of caption
p9 Fig 7 – Ti and Al are indistinguishable – please correct
p9 l201 – Wafer bonding is described as bringing new levels of “uniformity and control” but sacrificial release was described as “reliable”. Please clarify the difference between the two processes better as high “uniformity and control” may be equated with reliability.
p11 Fig 9 – Parts d and e are out of sequence with reference to the text. Either they should be used to create a separate figure (Fig 12) or it should be noted in the caption that they are included in Fig 9 to allow comparison with the other structures presented there.
p11 l263 – Correct “the dielectric constant and the use of” to “the dielectric constant, the use of”
p12 l276 – Replace “exciting” with “interesting”. It would also be helpful to provide a couple of references on the applications of flexible devices to link the fabrication issues to applications better.
p12 l280 – Replace “flavors” with “types”
p12 l287 – “Pillcam” is a trademark of Medtronic. It should be marked as such or a different term should be used.
p12 l290 – It is not clear at this point in the paper how the fabrication process described relates to capsule endoscopy (which is what a pillcam seems to provide). More detail should be provided or the explicit reference should be removed.
p13 l319 – 320 – The statement regarding active electronics within probes requires some clarification. For example, it is expected that, within 5 years, the majority of ultrasound systems sold will be handheld, where the probes will certainly contain electronics. Hence amendment to mention the direction of travel of ultrasound probes / systems is needed.
p13 l324 – Change “essentially strengthen” to “amplify”
p14 l328 – Please clarify what is meant by a “system channel” to which probe channels are matched in number.
p14 l329 – CMUT technology is certainly well-suited to integration with electronics. However, there are also numerous approaches to integrate piezoelectric devices with electronics, from PMUTs to “bulk” piezoelectrics, including piezocomposites and piezoceramic / single crystals. The work in Delft (de Jong / Pertijs) is an example as is the award winning work of Wildes and his colleagues at GE Schenectady and the work of Horsley at UC Davis. Interestingly, all these examples use different integration approaches. This should be explained and examples should be cited so as to present the reader with a true picture.
p14 l332 – The approach of Horsley cited previously also utilises monolithic integration. Hence this cannot be claimed to be “unique” to CMUTs. Please correct the text.
p14 l337 – Change “didn’t” to “did not”. Butterfly Network Inc. is a startup company that has only very recently begun selling systems; it is not clear if this is real “commercial traction” or only another transient. It would be helpful to cite other companies for comparison.
p15 l401 – It is not helpful to state that “array size [is]…severely limited due to size limitations”. The phrase “size limitations” should be changed to “application constraints such as blood vessel diameter”.
p17 l453 – Correct “pulsars” to “pulsers”.
p17 l454 – The 5 MHz frequency noted is very unusually high for HIFU applications which typically require high penetration into the body with low aberration. The reason for such a high maximum frequency, or at least its unusual nature, should be noted so as to guide the reader.
p17 l462 – Rephrase e.g. “As there have already been many applications of CMUTs, we cannot present them all here. Instead, we have selected applications of particular interest from our own work.”
p18 l479 – Accurate co-registration of MRI with a HIFU source is a pre-requisite for MR-guided focused ultrasound surgery (e.g. in systems from Profound Medical and InSightec). The same applies to US imaging. As such, it is misleading to state that “co-registration of the HIFU delivery location with the imaging [is] difficult”; this is an accepted part of both MRgFUS and USgFUS systems. The technology integration is not usually just as close as in the system described in this paper but systems (e.g. from Chongqing HAIFU) which have mechanically co-mounted ultrasound imaging arrays and HIFU sources have acceptable performance. This should be clarified in the paper, with appropriate references.
p20 l517 – Numerous ultrasound imaging systems are now available with a range of form factors for handheld operation. Clarius (USA) is an example of a supplier that provides a system using a mobile phone as the display and for communications; its system is even wireless(!) Healcerion (Korea) offers the same capability. CMUTs are certainly attractive in such situations but the bang! suggests exaggerated respects for what Butterfly has achieved. (It is noted that one of the authors is associated with Butterfly.) A more balanced view should be presented.
Additionally, a specific price is quoted. For a technical paper such as this one, there should be a statement of what this price represents in technical terms. It is unclear, for example, if it covers the costs of the seven years of development that has been required to date or if these costs have been written off in some way. This is important as it is recognised that the various components of piezoelectric probes represent a very large part of a low-cost system and are difficult to scale. Further, the quality of piezoelectric probes can be difficult to maintain and often requires very high sampling rates (up to 100%) during quality assurance. As this can lead to high costs, it would be helpful for the reader to understand the equivalent manufacturing yield and potential reduction in QA costs with CMUTs. Alternatively, the price can be removed from the paper.
p20 l519 – Philips announced their CMUT / PMUT foundry capability earlier than is stated. See https://www.innovationservices.philips.com/looking-expertise/mems-micro-devices/mems-applications/capacitive-micromachined-ultrasonic-transducers-cmut/# for more information. This website includes a presentation which can be downloaded and referenced formally. Foundry services are also beginning to emerge for PMUTs e.g. see https://www.mdpi.com/2504-3900/2/13/925/pdf. This should be mentioned and referenced to provide context.
p20 l529 – The bang! should also be removed from the end of this sentence. It is not clear what it signifies, whether 23 companies is a high total (in a developing field) or a very low total (given the large amount that has been achieved and the remarkable promise of CMUTs and PMUTs for integrated ultrasound solutions).
Reviewer 2 Report
The proposed paper entitled “Advances in Capacitive Micromachined Ultrasonic Transducers” is a review of the advances in CMUTs dealing with modeling, fabrication, integration, applications and industrializations.
If the overview presentation is quite clear, I would like to address a few issues which motivate in my opinion a revision of the article:
- Too many topics (modeling, fabrication, integration, applications and industrializations) are considered in the article and as a consequence, many of thems are very superficially discussed.
o For example, many approaches in modeling are not covered ; particularly about the nonlinear behaviour and the acousto-mechanical coupling (different original works achieved by the Greman Laboratory (Tours, France), the Bilkent University (Bilkent, Turkey), IMEC (Leuven, Belgium), or Department of Electrical Engineering and Computer Science (University of Liège, Belgium).
o About the fabrication, the authors propose a whole list of processes without considering and comparing the geometries and the acoustical performances.
- The review article is maybe too much focused on the works of the Khuri-Yakub Group of the Stanford University. In my opinion, this is not a general review article.
o For example, in the first 29 bibliographic references (up to page 11), there are 21 concerning this group. Overall, there are 33/66 bibliographic references of this group.
o Another point concern the applications of CMUTs. Moreover, the authors are clear on this subject, since they write in the paragraph devoted to the applications: “… . We present just a handful of the most recent projects carried out mostly by our group. …”. It is regrettable that other potential applications like Structural Health Monitoring, Detection of internal solid material defects using Lamb waves, Biometric technology and capability, and so on … are not presented.
o Lastly, only three figures are referenced (7, 16 and 17) and each time, a member of the Khuri-Yakub group is one of the concerned authors. This means that all the figures relate to works of this group.
- Several mistakes / misprints in the equations are noticed. I examine in details these points in the following.
In the following, I report a few points to be revised in details.
p. 3:
- About the equation (3):
o The electrostatic force relative to the x-axis on the figure 2 have to be positive (as the acoustic pressure). Moreover, the coefficient “1/2” is omitted in the expression of the electrostatic force.
- About the equation (4):
o The right-hand side member of the equation (4) is erroneous : -Vac(t)2 à +Vac(t)Vdc
- About the equation (5):
o Vdc is not time-dependent : Vdc(t) à Vdc
p. 4:
- About the equation (8):
o In the second expression : E0 à C0
p. 5:
- “For a piston transducer of radius a …” à “For a circular piston transducer of radius a …”.
o Previously, it is written that it is possible “to derive the impedance of plates of various shapes such as circular, rectangular, …”. In the same way, some works deal with the calculation of the acoustic impedance of flexible circular and rectangular plates with different clamping conditions.
- About the equation (11):
o The expression of the angular frequency is erroneous: the stiffness constants have to be in the numerator and the mass constants in the denominator.
- About the equation (12) and particularly the expression of X (w):
o The coefficient “cac” is not defined in the text.
- About the paragraph §2.3 “Finite Element Modeling”:
o The presentation of the finite element method is required but the explanation in details of this very well-known method is not relevant in this context (from “The underlying physics …” to “… and the true solution”). As previously discussed, it would be more interesting to present other original modeling approaches.
Round 2
Reviewer 1 Report
Clearly, the authors have conscientiously considered all the points made under review, whether they have corrected them or not.
Reviewer 2 Report
Overall, I appreciate the explanations of the authors considering my queries about the particular focus on the works of the Khuri-Yakub Group.
On the one hand, some references have been added to present other approaches and on the other hand, the authors clarify the specific context of the paper which is an invited one in a particular issue about the biomedical imaging applications.
Moreover, several mistake / misprints, especially in the equations, have been corrected.
Answers to the reviewer 1 comments are also very relevant and the consequential modifications in the paper significantly improve its readability.
Therefore, I accept the paper for publication.
As requested by the authors, I bring to their attention some references in particular about the CMUT modelling approaches (finite difference models, multiharmonic finite element formulations, mode-displacement methods):
C. Meynier, F. Teston and D. Certon, “A multiscale model for array of capacitive micromachined ultrasonic transducers”, Journal of Acoustic Society of America,Vol. 128, n° 5, pp. 2549-2561, 2010.
M. Berthillier, P. Le Moal and J. Lardiès, “Dynamic and acoustic modelling of capacitive micromachined ultrasonic transducers”, IEEE International Ultrasonics Symposium, Orlando (USA), October 18-21, 2011.
N. Sénégond, A. Boulmé, C. Plag, F. Teston and D. Certon, “Fast-time domain modelling of fluid-coupled CMUT cells:From the single cell to 1-D linear array element”, IEEE Transactions on Ultrasonics, Ferroelectrics, and Frequency Control, Vol. 60, n° 7, pp. 1505-1518, 2013.
A. Halbach and C. Geuzaine, “Steady-state, nonlinear analysis of large arrays of electrically actuated micromembranes vibrating in a fluid”, Engineering with Computers, Vol. 34, n° 3, pp. 591-602, 2018.
To conclude, I have just noted a few typing errors:
- p. 3 : “Vacuum” à”vacuum”
- p.8 – l. 187 : “It” à “it”
- p.10 – l. 238 “thw” à “the”
and maybe, it will be more accurate p. 18 – l. 479 to specify :
- “selected applications” à “selected biomedical imaging applications”
